# Surface Characteristics and Hydrophobicity of Ni-Ti Alloy through Magnetic Mixed Electrical Discharge Machining

**DOI:** 10.3390/ma12030388

**Published:** 2019-01-26

**Authors:** C.C. Feng, L. Li, C.S. Zhang, G.M. Zheng, X. Bai, Z.W. Niu

**Affiliations:** 1School of Mechanical Engineering, Shandong University of Technology, Zibo 255000, China; zy94fcc@163.com (C.C.F.); zhengguangming@sdut.edu.cn (G.M.Z.); cllyli1109@163.com (X.B.); niuzongwei@sdut.edu.cn (Z.W.N.); 2School of Mechanical Engineering, Zibo Vocational Institute, Zibo 255000, China; sdutwd@163.com

**Keywords:** Ni-Ti alloy, surface characteristics, hydrophobic, magnetic mixed EDM

## Abstract

Nickel–titanium (Ni-Ti) alloy has been selected as stent material given its good biocompatibility. In this study, experimental research on this material was conducted using magnetic field-assisted electrical discharge machining (EDM). The surface topography of the machined workpiece was analyzed with a scanning electron microscope (SEM). Hydrophobicity was measured by using an optical contact angle measuring instrument. The roughness values of different positions on the surface were measured using a TR200 roughness instrument. Results showed that the composite structure of solidification bulge–crater–pore–particle can be prepared on the surface of the Ni-Ti alloy through magnetic mixed EDM using suitable processing parameters. Moreover, the contact angle of the surface reaches 138.2°.

## 1. Introduction

The element composition of medical Ni-Ti alloys generally suggests a minimal difference between nickel and titanium contents and the presence of only trace impurities. Ni-Ti alloy has been used in biomedicine given its good shape-memory property, superelasticity, and excellent corrosion resistance [1,2,3]. It is also an ideal material for vascular stents considering its excellent in-body deformation and high strength. Among vascular stents, porous drug-eluting stents use surface micropores, which have the efficacy of drug-eluting stents and the long-term safety of bare metal stents for drug storage and release, thereby showing attractive prospects [4]. As a biological implant, stents must demonstrate good blood compatibility, which is closely related to the hydrophobicity of the material surface. The improvement of materials’ hydrophobicity can reduce the adhesion and activation of platelets on the material surface [5,6]. Simultaneously, the good hydrophobicity of the surface can also achieve a slow release of the drug and prolong its action time.

Ni-Ti alloy has the characteristics of low thermal conductivity, high ductility, and high viscosity. The use of traditional machining methods causes high tool wear and work hardening, resulting in the workpiece’s poor surface quality [7,8]. Electrical discharge machining (EDM) is based on electrical corrosion to corrode a material [9,10]. EDM’s optimal advantage is that it can efficiently process any conductive materials, regardless of their mechanical properties. It is only related to the electrical conductivity and thermal properties of materials [11,12]. At present, few studies are available on preparing multi-scale microporous and hydrophobic surfaces on a metal substrate through EDM. Researchers have found that a surface has good hydrophobic properties when its structural characteristics are composed of craters, bulges, micropores, and particles [13,14]. A new process of preparing hydrophobic surfaces with micropores through EDM is proposed in the present study on the basis of previous studies that have shown that the surface of Ni-Ti alloy machined through EDM is mostly composed of craters and bulges [15].

When a gas-rich working medium flows through the processing area, part of the gas dissolves into the molten metal to form pores because additional magnetic mixing allows additional air to mix into the working medium. Many pores are found on the surface of scaffolds, thereby enabling these scaffolds to have a high drug loading and improved hydrophobicity. A magnetic stirring device was added to the EDM machine to improve the microporous characteristics of the processed Ni-Ti alloy surface. The effects of different parameters on the surface characteristics were investigated, and the effect of surface morphology on the surface hydrophobicity of Ni-Ti alloy was analyzed.

## 2. Experimental Procedure

### 2.1. Equipment and Materials

The experimental device was modified on a DM71 (DM71, Changde Ltd., Taizhou, China) precision EDM machine. The magnetic mixed device was installed on the machine table. The fluid container was placed on the magnetic mixed device, thus quickly stirring the working fluid. The workpiece was fixed onto the workpiece fixture, whereas the tool electrode was fixed onto the tool fixture and then connected with the spindle. Figure 1 illustrates the structural principle.

A special spark oil was selected as the working medium, and its index is listed in Table 1. The workpiece material was Ni-Ti alloy, and its chemical composition is presented in Table 2. The tool electrode material was red copper with the dimensions of *Φ* 16 mm × 150 mm. Table 3 displays the thermal and physical properties of copper and Ni-Ti alloy. In Table 3, both materials exhibit a minimal difference in density and melting point, but the thermal conductivity and specific heat capacity are much higher in copper than in Ni-Ti alloy. This observation indicates that Ni-Ti alloy has poor heat dissipation, and the same heat can persist on its surface for an extended period.

### 2.2. Experimental Parameters

The magnetic stirring device had a power of 30 W and a rotating speed of 2600 r/min. The effects of peak current and pulse duration on surface microstructure, surface roughness, and static contact angle were investigated. The EDM parameters were as follows: the peak currents were 1.5, 4.5, and 9 A; the pulse duration was 15, 30, 60, and 90 µs; the voltage was 50 V; and the pulse interval was 5 µs.

### 2.3. Sample Testing and Characterization

The samples were ultrasonically cleaned for 8 min, followed by air drying, and were examined using a scanning electron microscope (SEM; Apreo, FEI Ltd., Hillsboro, OR, USA) and energy dispersive spectroscopy (EDS; Apreo, FEI Ltd., Hillsboro, OR, USA) to analyze the surface topography and elemental composition, respectively. The parameters of EDS were kv: 20, mag: 2000, takeoff: 36.1, live time(s): 26.7, and amp time (µs): 7.68. The EDS adopts area analysis for the machined surface. X-ray diffraction (XRD; D8 Venture, Bruker Ltd., Madison, WI, USA) analysis was used to analyze the surface compounds. The static contact angle of the surface was measured using a video optical contact angle measuring instrument (OCA15EC, Dataphysics Ltd., Stuttgart, Germany), and the volume of the water droplet was 5 µL. Five positions on the surface were selected to measure the contact angle, and the average value of the measurement results was obtained. A roughness meter (TR200, Times Group Ltd., Shandong, China) was used to measure roughness. Five workpiece positions were selected for measurement, and then the average value was determined. The samples were mounted, polished, ultrasonically cleaned, and etched for a subsurface microstructure observation using the SEM. The etchant consisted of 10 ml HF, 40 ml HNO_3_, and 50 ml deionized water.

## 3. Results and Analysis

### 3.1. Effect of Parameters on Surface Morphology

Two sets of data were selected for the EDM of Ni-Ti alloy without being magnetically mixed to highlight clearly the role of magnetic mixing. The results are presented in Figure 2. Figure 2a shows very few holes on the surface when the current is 1.5 A and the pulse duration is 60 µs. No stomatal feature on the surface of Figure 2b is observed when the current is 9 A and the pulse duration is 30 µs.

The surface micrographs of Ni-Ti alloy after magnetic mixed EDM under different processing parameters are different. The surface contains a different number of pores under most processing parameters. The formation of pores is due to a large amount of heat generated during the EDM of Ni-Ti alloy, which causes the material to enter a molten state. At that time, gas dissolves into the molten metal. The molten metal quickly cools and solidifies with the cooling flow of the working fluid, and then some of the gas bubbles are expelled with the splash of the molten material. The unexpelled gas is trapped in the re-solidified material, resulting in micropores [16]. The gas is mainly produced in two steps. The first step occurs when the working fluid is quickly stirred using the magnetic stirring device, which enables air to enter the liquid and the working fluid becomes rich in gas. When the gas-carrying working fluid flows through the molten metal surface, the gas enters the molten metal. The other step is when the insulating medium is ionized to produce hydrions during the EDM’s discharge process. The hydrions in the discharge channel are reduced on the negative polarity surface to produce hydrogen gas. Furthermore, a certain amount of working fluids evaporates at a high temperature to form water vapor, while the liquid produces oxygen due to pyrolysis at a high temperature. These gases also dissolve into the molten metal. Moreover, the velocity of the liquid passing through the processing area is 1.8 m/s. Rapid cooling prevents the gas from overflowing in large quantities from molten metals. Thus, the stomatal characteristics on the surface increase. However, the formation of pore morphology is due to a large amount of gas mixed by magnetic stirring because, in the absence of magnetic stirring, and even with the same process parameters, only an individual pore was found on the surface [17]. Figure 2 shows that, when magnetic mixing is not used, only an individual pore is obtained, or no pore exists on the surface.

Figure 3a–c indicates that the surface of Figure 3a contains many shallow pores. The result is that the parameter of the peak current is 1.5 A and the pulse on time is 15 µs. A short discharge time indicates a short time of gas entering the molten metal, resulting in a minimal gas entry. In addition, the discharge pulse energy is small, and the pressure is inadequate to remove numerous materials. Thus, additional molten materials are re-solidified on the surface. The surface forms numerous shallow pores, and several solidification bulges are connected by craters. When the pulse duration increases to 30 µs, the number of pores does not increase significantly, whereas the craters between the solidification bulges are clearly enlarged to form gaps, and several spherical fragments are found in the gap. This phenomenon occurs because, when the peak current is 1.5 A, the pulse energy increases with the pulse duration. The gap between the solidification bulges expands and occupies the position of the pores, and the pores do not increase considerably. Spherical debris is formed by re-solidifying the material on the surface of the workpiece when it meets the flowing working fluid after gasification. When the pulse duration is further extended to 60 µs, the number of pores does not increase, but the gap between the solidification bulges is further enlarged. Moreover, the spherical debris in the gap develops into a “coral reef” structure. This phenomenon occurs due to the continuous increase in pulse duration, which expands the discharge channel and leads to low discharge energy, and most of the molten materials remain on the surface, rather than being splashed. The continuously expanding gap and “coral reef” morphology further occupy the position of the pores. Therefore, although the pulse duration increases, the number of pores does not increase considerably. The “coral reef” is formed by the re-solidified materials with random irregular debris, and the debris is re-solidified from the splashed molten material or the vaporized material through rapid dielectric quenching [18].

Figure 3d–f illustrates that, when the pulse duration was 15 µs, the SEM found 14 holes on its surface. These holes are due to the short discharge time and the minimal amount of gas to enter the molten metal. In comparison with Figure 3a, the discharge pulse energy is larger, and the molten material is removed. Thus, the number of shallow pores on its surface reduces. When the pulse duration increases to 30 µs, the material removal rate increases with the continuous increment in pulse discharge energy, and the shallow pores on the surface are invisible. Furthermore, given the increase in pulse duration, the gas has additional time to enter the molten metal, and the depth of the pore increases slightly. When the pulse duration further increases to 60 µs, the increase in pulse duration leads to additional gas that enters the molten metal, thereby forming deep pore characteristics on the surface.

Figure 3g–i demonstrates that craters and pores are nonexistent on the surface when the peak current is 9 A and the pulse duration is 15 µs. This result is due to the small pulse duration that leads to minimal gas entering the molten metal. Simultaneously, considering the high discharge pulse energy, the molten metal has been considerably removed before forming pores. Thus, no pore characteristic is observed on the surface. This trend can be observed from the changes in Figure 3a,d. When pulse duration increases to 30 µs, many deep pores are found on the surface, thereby increasing time and the quantity of gas that enters the molten metal due to the increase in pulse duration. In comparison with Figure 3e, the discharge pulse energy increases, resulting in the removal of the shallow surface pore characteristics. Thus, the surface pore presents a deep state. When the pulse width increases to 60 µs, additional gas enters the molten metal. The shallow pores on the surface are removed considerably, and the depth of the pores on the surface is deeper than that of Figure 3h considering the large discharge pulse energy.

### 3.2. Effect of Surface Topography on Contact Angle

Wettability is an important property of a solid surface. The wettability of a material surface is typically measured by the static contact angle θ. That is, θ > 90° are hydrophobic surfaces, whereas θ < 90° are hydrophilic surfaces. The intrinsic contact angle of Ni-Ti alloy is 70°, and the surface exhibits a hydrophilic characteristic. An appropriate surface morphology must be constructed to improve the contact angle of Ni-Ti alloy and obtain improved hydrophobicity.

Figure 4 shows a remarkable difference in the contact angles of different surface morphologies. When the pulse duration is 60 µs and the peak current is 1.5 and 4.5 A, the surface contact angles of the two parameters are greater than 130°, and the size of the solidification bulges on their surfaces is large. The solidification bulges exhibited in Figure 4c are more regular and denser than those of Figure f. Therefore, when the droplets come into contact with the surface, the bulges form support points to lift the droplets, and the gap between the bulges closes considerable air. The difference in pressure prevents the droplet from fully entering the gap between the solidification bulges, thereby resulting in the formation of an “air cushion” effect, which reduces the contact between the droplet and the solid [19,20,21]. Furthermore, Figure 4c displays that the “coral reef” structure between the solidification bulges is not in a single dimension. In addition, the droplets that enter the gap form the secondary “air cushion” effect on the “coral reef” structure, and the surface of the material has changed from hydrophilicity to hydrophobicity. The contact angle presented in Figure 4f is slightly smaller because no clear demarcation line is found between the solidification bulges on the surface, and no special morphology between the bulges is observed. However, a certain number of pores are found on the surface. In addition, Figure 3f illustrates that the pore size of the surface is small and has a certain depth. Thus, when the droplet is on its surface, the bulges form support points to lift the droplets, and the droplet cannot enter into the micropore completely given the difference in pressure. The “air cushion” effect is formed and transformed into a hydrophobic surface.

The surface morphology depicted in Figure 4a,b is similar to that of Figure 4c, but the size of the solidification bulges increases with the pulse duration. The solidification bulges in Figure 4a,b are smaller. When a droplet is acted upon by gravity, the contact area is small and the pressure is high. Thus, small solidification bulges can pierce the droplet and lead to a large area in contact with the droplet and therefore a small contact angle.

Figure 4d,e shows that the contact angle of the surface with several shallow pores is below 90°; this surface is not hydrophobic. This condition is due to several shallow pores on their surfaces, thereby enabling droplets to easily enter the shallow pores on this type of surface given gravity and capillary forces [22]. Therefore, the contact area between the droplet and the solid increases, and catching sufficient air to form the pressure difference is not feasible; thus, no “air cushion” effect is observed [19,20,21]. The contact angle is small considering the large contact surface between the droplet and the solid. However, given the change in the surface microstructure, the contact angle remains slightly improved in comparison with the intrinsic contact angle. In addition, the surface morphology depicted in Figure 4d changes minimally, that is, neither the large size of the solidification bulges nor the formation of a large number of shallow pores; therefore, the contact angle increases slightly. When the peak current is 4.5 A, the pulse duration is 15 and 30 µs, and the surface morphology contains numerous shallow pores. Thus, their contact angles are nearly the same, and the surface remains hydrophilic. However, when the pulse width is 60 µs, many deep pores are formed, and the contact angle is considerably increased by the “air cushion” effect.

When the peak current and pulse duration are 9 A and 15 µs, respectively, Figure 4g shows the formation of large solidification bulges, but these bulges are only disorderly overlapping. No boundaries are found between these bulges, and no pores are formed on the surface. When the droplet is on its surface, the height of the convex overlap is different, thus leading to the different heights of the bulges that support the droplets, and the surface is only slightly hydrophobic. With the increase in the pulse duration, the surface morphology also changes considerably. Figure 4h,i indicates similar surface morphologies, but their contact angles differ remarkably because, when the pulse duration is 30 µs, the surface morphology contains additional shallow pores. Moreover, the droplets will be in contact with additional solids, and the contact angle will be smaller than the one displayed in Figure 4g. When the pulse duration increases to 60 µs, the depth of the pores deepens, and the local “air cushion” effect occurs. The contact angle is larger than the one presented in Figure 4g.

Figure 3 shows that the number of pores on the surface increases when the peak currents are 4.5 and 9 A given the increase in pulse duration. Figure 4 illustrates that, when the peak currents are 1.5 and 4.5 A, the hydrophobicity of the surface increases with the pulse duration. Considering that the peak current is less than 1.5 A, the discharge becomes atypical, and the processing efficiency is extremely low. Only the pulse duration is considered to continuously increase to observe the change in surface characteristics. When the pulse duration is increased to 90 µs, the results obtained are depicted in Figure 5. Figure 5b demonstrates a high multiple surface topography under the peak current of 1.5 A.

Figure 5 shows that, when the pulse duration increases to 90 µs, the contact angles of the three currents are reduced. When the peak current and pulse duration are 1.5 A and 90 µs, respectively, the size is slightly larger in the solidification bulges than with the pulse duration of 60 µs. However, Figure 5b displays no “coral reef” structure on the surface. Therefore, a large solidification bulge indicates a large contact area. Furthermore, considering that no “coral reef” structure is found on the surface, the secondary “air cushion” effect cannot be formed, thereby reducing the contact angle. When the peak current is 4.5 A, additional craters are found on the surface. The surface morphology is poor, and the air cushion effect cannot be formed. Thus, the contact angle decreases considerably. When the peak current is 9 A, irregular discharge traces are found on the surface and the number of pores decreases. Thus, the “air cushion” effect is reduced, and the contact angle is also reduced.

### 3.3. Element Analysis and XRD Analysis of Workpiece Surface

The analysis presented in Section 3.2 suggests that the contact angles of different surface morphologies are considerably different. Thus, the two samples with the largest difference in contact angles were selected for surface element analysis. The analysis area is illustrated in the white box in Figure 3c,h, and the results are presented in Figure 6.

The very high temperature of plasma instantly melted and vaporized the material during the discharge process. The copper electrode and Ni-Ti workpiece were eroded, and complex chemical reactions between the vaporized gas and the molten pool were observed. Therefore, the existence of Cu was found on the surface of the workpiece. As a result of magnetic mixing, the high cooling rate of liquid dielectric resulted in Cu diffusion on the surface, rather than it being flushed away. The presence of O can be attributed to decomposing water in the discharge gap. Figure 6 shows that different surface topographies have various C contents, and the surface morphology exhibited in Figure 6b contains “coral reef” with the highest C contents on the surface. This result is due to its processing parameters, *I* = 1.5 A and *T_on_* = 60 µs, and to the large pulse duration, thereby leading to expanding discharge channels. The corrosion products are not thrown out in time but are rather re-solidified on the surface. Simultaneously, given the poor thermal conductivity and extensive pulse width of Ni-Ti alloy, the working fluid decomposes considerable C for the highly active Ti to absorb the considerable C in the process of re-solidification. Thus, the C content on the surface is high. Figure 6a,b indicates that the Ni content is different under various processing parameters because Ni has a lower evaporation temperature than Ti. Moreover, with the increase in the pulse duration, additional Ni is removed from the surface by the liquid dielectric. In addition, TiC substances are found on the surface of Figure 6c,d, which will further decrease the content of Ni. Thus, the Ni content decreases.

Ni-Ti alloy has good biocompatibility and corrosion resistance considering the natural titanium film found on its surface. However, the natural titanium film cannot withstand the long-term erosion of human physiological body fluid and blood. Once the natural titanium film breaks down, nickel ions will precipitate, and excessive nickel ions will cause anaphylaxis and tissue necrosis [23]. The TiC prepared on the surface of Ni-Ti alloy can effectively improve its biocompatibility and mechanical strength, and inhibit nickel ion precipitation [24]. Figure 6c,d presents the XRD of two characteristic surfaces. TiC strengthening phases are found on the machined surface, thereby confirming that C on the workpiece surface exists in the form of a compound. The positions of TiC diffraction peaks in the two graphs are the same but the intensity is different, indicating that the content of TiC varies.

### 3.4. Subsurface Microstructure

Two specimens with the largest difference in contact angle were selected for the cross-section analysis. The results are shown in Figure 7.

The recast layer depicted in Figure 7b is non-uniform, but this layer is thicker than the one shown in Figure 7a because Figure 7b exhibits a large pulse duration. Moreover, when the duration of discharge is extended, melting isothermals penetrate further into the material interior, thereby resulting in the molten zone’s further extension into the material and a large recast layer thickness. The surface topography is displayed in Figure 3c. This layer is composed of solidification bulges, “coral reef,” micropores, and craters. Thus, the thick part of the recast layer is the solidification bulge, and the thinner part is the “coral reef.” The contact angle can be considerably increased because this unique structure can seal considerable air and form an “air cushion” effect. Figure 7a indicates that numerous materials are removed given the high pulse discharge energy, and the cross-section morphology is uniform and cannot form the “air cushion” effect.

### 3.5. Relationship of Surface Roughness and Contact Angle

Figure 8a illustrates the influence of processing parameters on the contact angle. When the peak currents are 1.5 and 4.5 A, the contact angle increases first and then decreases with the increase in pulse duration. According to the analysis in Section 3.2, this phenomenon occurs because the surface morphology has dramatically changed with the increase in pulse duration. When the pulse duration is below 60 µs, the size of the solidification bulges increases with the pulse duration, but the surface that forms the “air cushion” effect has not formed or is insufficiently large. This phenomenon leads to a large contact area between liquids and solids so that the contact angle increases are insufficiently large. Although the size of the solidification bulges is large when the pulse duration increases to 90 µs, the contact angle decreases differently with the decrease or absence of the “air cushion” effect. When the peak current is 9 A, the contact angle varies irregularly with the increase in pulse duration. However, when the pulse durations are 60 and 90 µs, the law is consistent with the law of other parameters because the suitable surface can form an “air cushion” effect, which helps enhance the contact angle. Within the range of parameters used in this study, the optimum parameters for preparing a hydrophobic surface are *I* = 1.5 A, *T*_on_ = 60 µs, and *V* = 50 V. However, considering the number of pores on the surface to facilitate drug loading, the parameters are *I* = 4.5 A, *T*_on_ = 60 µs, and *V* = 50 V.

Figure 8b depicts the effect of processing parameters on surface roughness. The pulse energy enlarges with the increase in the pulse duration and peak current. Thus, surface roughness increases accordingly. Several researchers have suggested that a contact angle increases with roughness [25]. However, Figure 8a,b shows that, when the pulse duration is between 60 and 90 µs, the surface roughness increases with the pulse duration, but the contact angle decreases with the increase in the pulse duration. Under the same pulse duration, the peak current and the roughness of the surface are large, but the contact angle is not necessarily large. The experimental results show that surface contact angles are not entirely increasing with surface roughness. Therefore, only suitable surface roughness and surface topography can achieve improved hydrophobicity.

## 4. Conclusions

This study investigated the processing of Ni-Ti alloy through magnetic mixed EDM with different parameters and carried out an analysis of its surface properties. The key results are summarized as follows.

(1) When the peak currents are 4.5 and 9 A and the pulse durations are 30 and 60 µs, the porous surface can be prepared on the surface of Ni-Ti alloy through magnetic mixed EDM. The composite structure of solidification bulge–crater–pore–particle can be prepared on the surface of Ni-Ti alloy through magnetic mixed EDM with suitable processing parameters.

(2) The processing parameters considerably affect the surface morphology. Different processing parameters can obtain various surface microstructures, and the surface microstructures directly affect the hydrophobicity of the surface. When the peak current and pulse duration are 1.5 A and 60 µs, respectively, the surface is composed of many large-sized solidification bulges and “coral reef” structures, and the contact angle of the surface can reach 138.2°. However, considering the number of pores on the surface to facilitate drug loading, the optimal parameters of this study are *I* = 4.5 A, *T*_on_ = 60 µs, and *V* = 50 V.

(3) The C content on the surface of the workpiece increases rapidly with the increase in the pulse width and the decrease in the peak current.

## Figures and Tables

**Figure 1 materials-12-00388-f001:**
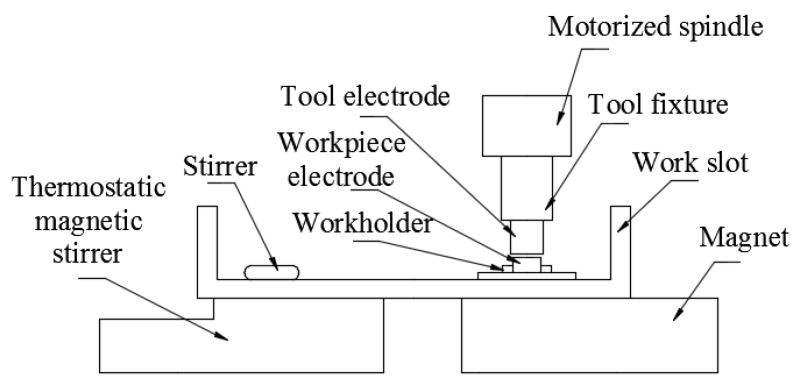
Schematic of the machine tool.

**Figure 2 materials-12-00388-f002:**
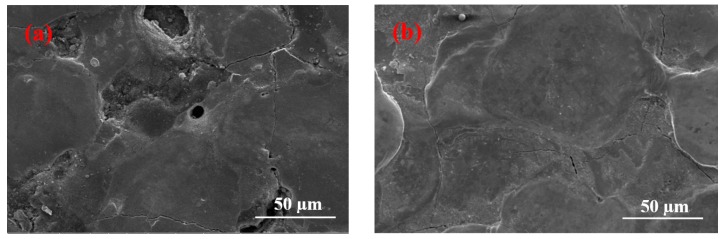
Surfaces machined by EDM without additional magnetic mixing. (**a**) *I* = 1.5 A, *T*_on_ = 60 µs, (**b**) *I* = 9 A, *T*_on_ = 30 µs.

**Figure 3 materials-12-00388-f003:**
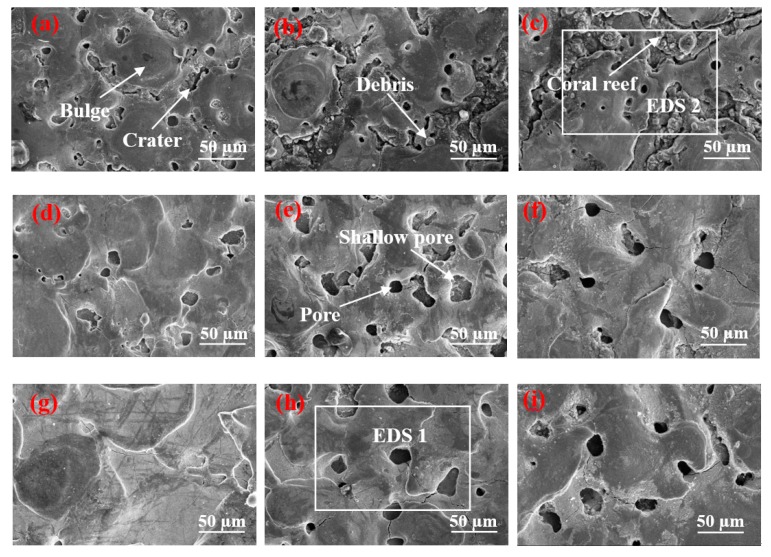
Surface morphology of surface at different processing parameters. (**a**) *I* = 1.5 A, *T*_on_ = 15 µs; (**b**) *I* = 1.5 A, *T*_on_ = 30 µs; (**c**) *I* = 1.5 A, *T*_on_ = 60 µs; (**d**) *I* = 4.5 A, *T*_on_ = 15 µs; (**e**) *I* = 4.5 A, *T*_on_ = 30 µs; (**f**) *I* = 4.5 A, *T*_on_ = 60 µs; (**g**) *I* = 9 A, *T*_on_ = 15 µs; (**h**) *I* = 9 A, *T*_on_ = 30 µs; (**i**) *I* = 9 A, *T*_on_ = 60 µs.

**Figure 4 materials-12-00388-f004:**
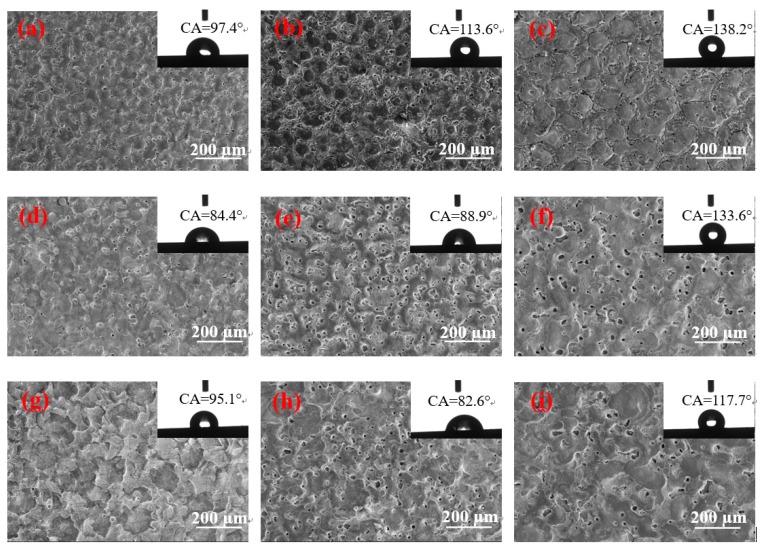
Surface morphology and contact angle under various processing parameters. (**a**) *I* = 1.5 A, *T*_on_ = 15 µs; (**b**) *I* = 1.5 A, *T*_on_ = 30 µs; (**c**) *I* = 1.5 A, *T*_on_ = 60 µs; (**d**) *I* = 4.5 A, *T*_on_ = 15 µs; (**e**) *I* = 4.5 A, *T*_on_ = 30 µs; (**f**) *I* = 4.5 A, *T*_on_ = 60 µs; (**g**) *I* = 9 A, *T*_on_ = 15 µs; (**h**) *I* = 9 A, *T*_on_ = 30 µs; (**i**) *I* = 9 A, *T*_on_ = 60 µs.

**Figure 5 materials-12-00388-f005:**
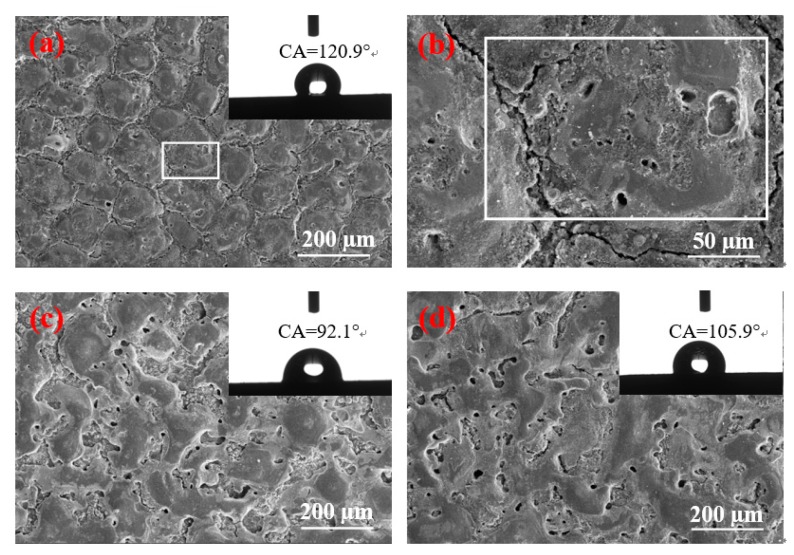
Surface topography and contact angle under different peak current conditions with pulse duration of 90 µs. (**a**,**b**) *I* = 1.5 A, (**c**) *I* = 4.5 A, (**d**) *I* = 9 A.

**Figure 6 materials-12-00388-f006:**
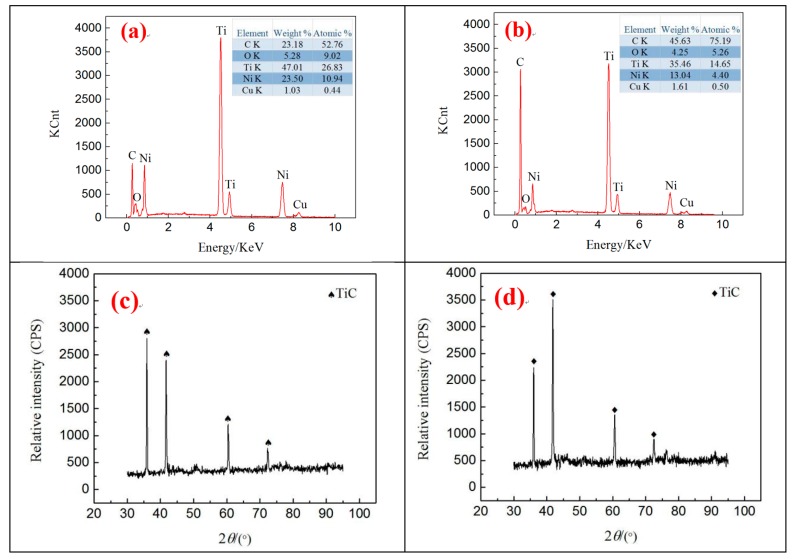
Surface element analysis and XRD analysis under different processing parameters. (**a**) *I* = 9 A, *T*_on_ = 30 µs; (**b**) *I* = 1.5 A, *T*_on_ = 60 µs; (**c**) *I* = 9 A, *T*_on_ = 30 µs; (**d**) *I* = 1.5 A, *T*_on_ = 60 µs.

**Figure 7 materials-12-00388-f007:**
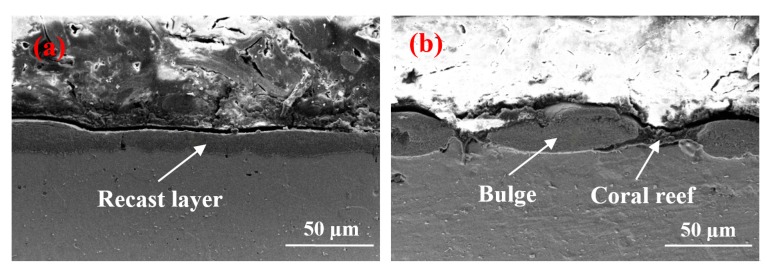
Cross-sectional view of recast layer. (**a**) *I* = 9 A, *T*_on_ = 30 µs; (**b**) *I* = 1.5 A, *T*_on_ = 60 µs.

**Figure 8 materials-12-00388-f008:**
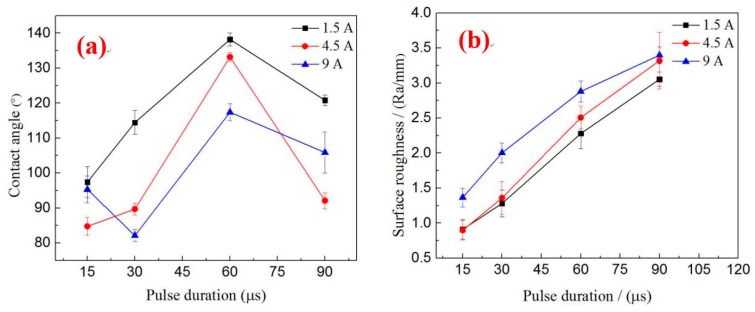
Effect of machining parameters on contact angle and surface roughness. (**a**) Influence of parameters on contact angle; (**b**) Influence of parameters on roughness.

**Table 1 materials-12-00388-t001:** Index of special electrical discharge machining (EDM) oil.

Index	Viscosity (40 °C)/(mm^2^·s^−1^)	Density (20 °C)/(kg·m^−3^)	Aromatic Content	Pour Point/°C
Typical data	2.20	0.79	≤0.08	−10

**Table 2 materials-12-00388-t002:** Chemical composition of Ni-Ti alloy.

Chemical Composition	Ni	Ti	C	O	H
Content/%	≤50.9	>48	≤0.05	≤0.05	≤0.003

**Table 3 materials-12-00388-t003:** Physical properties of materials.

Materials	Density/(g·cm^−3^)	Melting Point/°C	Thermal Conductivity/(W·m^−1^·k^−1^)	Specific Heat/(J·kg^−1^·°C)
Copper	8.96	1083	383.3	410
Ni-Ti	6.45	1310	10	0.32

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
