# Peer review of "Surface Characteristics and Hydrophobicity of Ni-Ti Alloy through Magnetic Mixed Electrical Discharge Machining"

_materials, 2019, doi:10.3390/ma12030388_

Round 1

Reviewer 1 Report

In the reviewed manuscript authors discuss the results of their works on surface changes and hydrophobicity of Ni-Ti alloy by electrical discharge machining. Results of carried out studies revealed that the surface modification of Ni-Ti alloy substrates by magnetic field assisted EDM allowed the control of the surface composition, roughness, and its hydrophobicity.

The paper requires the following amendments:

1.    Abstarct: The EDM acronym must be defined.

2.    The novelty aspects should be highlighted in the final part of the Introduction.

3.    The surface composition in this case is important for its properties, so it should be also confirmed  by XPS studies. The XPS is the more accurate than EDS.

In my opinion, the paper may be recommended for publication in the Materials, after minor alterations.

Author Response

Special thanks to you for your good comments.

 Abstarct: The EDM acronym must be defined.

Response: The EDM in the abstract is extended to electric discharge machining and is defined in the abstract.

2. The novelty aspects should be highlighted in the final part of the Introduction.

Response: Novelty has been emphasized in the last part of the introduction. The amendments are as follows, “When a gas-rich working medium flows through the processing area, part of the gas will dissolve into the molten metal to form pores because additional magnetic mixed can allow additional air to mix into the working medium. Many pores are found on the surface of scaffolds, thereby enabling these scaffolds to have a high drug loading and improved hydrophobicity”.

3. The surface composition in this case is important for its properties, so it should be also confirmed by XPS studies. The XPS is the more accurate than EDS.

Response: In Section 3.3 of the revised draft, X-ray diffraction analysis was added to the analysis. TiC formation was found on the surface by XRD, which further indicates that C element is not C deposit. See Section 3.3 of the revised draft for details.

Special thanks to you for your good comments again.

Reviewer 2 Report

1. Some writing  mistake must be corrected such as:

line 30, 31, 101.... workpiece[..., material[, ...etc.

line 229, 230, 269, 276....1.5A, 4.5A...etc.

2. In the experimental part data about used dielectric fluid (deionized water) is missing. It is important concerning Section 3.3.

3. In micrographs (Figure 2, Figure 3, Figure 4) scale bar must be emphasized. Also, every micrograph must be marked with a related letter (a), (b), (c), (d).

4. Although authors chose 1.5 A at 60 s as optimum related to the hydrophobicity of the surface, surface element analysis (in Figure 5) prepared at this condition were not presented.

5. Authors conclusion concerning the relation  between the number of pores and facilitation for drug loading are not supported by experiment, argument or discussion.

Author Response

Special thanks to you for your good comments.

1. Some writing mistake must be corrected such as: line 30, 31, 101.... workpiece[..., material[, ...etc. line 229, 230, 269, 276....1.5A, 4.5A...etc.

Response: we are very sorry for our negligence of writing mistake. We have made corresponding corrections: line 32, 33, 118, 228, 229, 298 and 305. Workpiece [, material [, surface [, 1.5 A, 4.5 A, 1.5 A, 1.5 A, 4.5 A, 9 A.

2. In the experimental part data about used dielectric fluid (deionized water) is missing. It is important concerning Section 3.3. 

Response: The introduction of working medium has been added in section 2.1. The special spark oil is used in this experiment, and its index is shown in Table 1. This special spark oil contains a certain amount of water, so the O element in section 3.3 comes from the water in it.

3. In micrographs (Figure 2, Figure 3, Figure 4) scale bar must be emphasized. Also, every micrograph must be marked with a related letter (a), (b), (c), (d).

Response: We are very sorry for the loss of some content caused by the problems in uploading files. All images of missing scale bar and related letter have now been re-appended.

4. Although authors chose 1.5 A at 60 µs as optimum related to the hydrophobicity of the surface, surface element analysis (in Figure 5) prepared at this condition were not presented.

Response: Figure 5 becomes figure 6. I am very sorry that this may be a problem with file uploads, causing the content of Figure 6 (b) to be invisible. When the parameters are 1.5 A and 60 µs, figure 6(b) is its element analysis. At the same time, the reason for the high content of C element in figure 6(b) is also added. This result is due to its processing parameters are I=1.5 A and Ton=60 µs, and the pulse duration is large, thereby leading to expanding discharge channels. The corrosion products are not thrown out in time but re-solidified on the surface. Simultaneously, given the poor thermal conductivity and extensive pulse width of Ni-Ti alloy, the working fluid will decompose considerable C for the highly active Ti to absorb considerable C in the process of re-solidification. Thus, the C content on the surface is high.

5. Authors conclusion concerning the relation between the number of pores and facilitation for drug loading are not supported by experiment, argument or discussion. 

Response: Thank you very much for your suggestion. We can think about it from a different angle. Drug-eluting stents inhibit reactive intimal hyperplasia and reduce the occurrence of restenosis in later stage by the amount of drug loaded on the surface of stents. Porous drug-eluting stents have a large number of micro-holes on their surface. These micro-holes make the drug loading on the stent surface increase to a large extent, which will also make the stent work in the human body for a longer time. It is also mentioned in document 4 that surface micro- and nano-structures also give the stent more space to carry drugs and make the stent more effective in suppressing thrombosis.

Reviewer 3 Report

The paper consist in the study of the influence of different EDM parameters on the surface morphology (pores, roughness, bulges) and the resulting hydrophobicity of the material. The study aims at preparing multiscale microporous and hydrophobic surfaces on biocompatible metals by enhanced EDM process that has magnetic stirring device added to it.

Although the approach is interesting, the paper has major flaws and needs extensive work to make it publishable in a journal such as Materials from MDPI.

In general:

There are a lot of speculations in the discussions on the reason for more or less hydrophobicity, especially regarding the “air cushion” explanation which I think lacks support from literature and from experimental evidence. Extended discussion of the results compared to wettability experiments from textured surfaces study would be relevant. And coupling such discussion with the addition of line profiles, image of water droplet on the surface to see the area covered, etc… would help a lot!

I was also surprised by the absence of comparisons to any EDM machined surfaces without the magnetic stirring. The sole comparison is brief and done via the citation of only one reference: ref [17]. To make the point clear, the authors should have done and should have presented some tests without magnetic stirring, not necessarily in all conditions for I and T, but enough of them to make the point.

Here are some more specific comments:

-          a figure presenting a surface taken from SEM cannot tell anything about the water contact angle. We have to wait until the end of the paper to have values of water contact angles, which by the way are not given with their standard deviation. Idem regarding the roughness, it is essential to the study and only given at the end, again without any standard deviation.

-          Figures must have a scale so the reader can have an idea of the lateral size of the features presented and discussed. All images in the figures must have the letter that is associated to them in the caption (a, b, c, d, e, etc…) written on them or below so the reader can associate them easily.

-          In many places, the authors say the pores are shallow, bulges are bigger, etc… however, base on one SEM image it is not possible for the reader to be sure it is the case as SEM images and the reading we have from them is highly dependant on the contrast and brightness. Profiles of the surface must be presented to illustrate the discussion.

-          Regarding the number of pores, it would be better if some semi-quantitative numbers are given such as numbers of pores per a certain unit area. Authors are only employing expression such as “certain amount of shallow pores”.

-          As there are no scales on images, the reader have no idea of the relevancy of the magnification used to illustrate the results and to support the discussion

-          Clear definition of what the authors define as pores, bulges, craters, coral reef needs to be provided. A simple arrow on one SEM images would be sufficient.

Section 2.1: what is the working fluid used in the EDM process?

Section 2.3:

-          what are the EDS parameters for the chemical composition study

-          the authors say “ The samples were mounted, polished, ultrasonically cleaned, and etched for subsurface microstructure observation by SEM. The etchant consists of 10 ml HF [...]”. I have not seen any subsurface images, nor any discussion directly on data gathered from this subsurface study. If the images from Fig 1 to 4, are those images, then the discussion on the roughness, bulges, pores, etc. on wettability does not make sense because the images would not reflect the real surface. Regarding the HF, there is only fluorine mentioned in the conclusion… and it is in relation with contact angle. That is why I have doubts on the images presented and the relevancy of the discussion (all section 3)

Section 3.1 is highly focused on the gas entering the contact. But what about the cooling pattern/rate due to the circulation of the liquid?

Section 3.2:

-          the discussion regarding the Cassie & Baxter theory needs to be developed. Moreover, they focused on rather well defined and regularly patterned surfaces rather than randomly patterned surfaces that are closer to what the authors present.

-          “This is because there are some shallow pores on their surfaces, which results in droplets easily entering the shallow pores on this type of surface due to gravity and capillary forces.” References are needed to support such a claim.

-          “Therefore, the contact area between the droplet and the solid increases, and it is impossible to catch enough air to form the pressure difference, so there is no "air cushion" effect.” References are needed to support such a claim, or extend the discussion to prove it further.

-          What do the authors mean by “disorderly overlapping”?

Section 3.3:

-          The authors selected only two surfaces to conduct elemental analysis. I would be cautious on that as wettability is impacted by a combination of things: cleanliness, material microstructure and composition, surface morphology, size of the drop, etc. The authors must give precisions on where the analysis is performed. Are the spectrum presented in Fig5 obtained from maps, single point analysis, line profiles analysis, etc.? If it is single points, authors must show the surface and where the spectrum is taken. It is extremely important because the surface is heterogeneous in morphology and can consequently be heterogeneous in its composition as well.

-          On the spectrum in Fig5, why are the % presented for Ni are based on the L ray and not the K ray?

-          “This is due to Ni has lower evaporation temperature than Ti, and as the pulse duration increases, more Ni element is taken away from the surface by the liquid dielectric, so the content of Ni decrease.” Okay, but what about the possibility of creating a thicker layer of mixed liquid and metal residue that would consequently lower the concentration of Ni.

Author Response

Dear Reviewer:

Thank you for your letter and for the reviewer’ comments concerning our manuscript entitled “Surface Characteristics and Hydrophobicity Analysis of Ni-Ti Alloy by Magnetic Mixed EDM” (ID: 412318). Those comments are all valuable and very helpful for revising and improving our paper, as well as the important guiding significance to our researches. We have studied comments carefully and have made correction which we hope meet with approval. Revised portion are marked in red in the paper. The main corrections in the paper and the responds to the reviewer’s comments are as flowing:

Responds to the reviewer’s comments:

1. There are a lot of speculations in the discussions on the reason for more or less hydrophobicity, especially regarding the “air cushion” explanation which I think lacks support from literature and from experimental evidence. Extended discussion of the results compared to wettability experiments from textured surfaces study would be relevant. And coupling such discussion with the addition of line profiles, image of water droplet on the surface to see the area covered, etc… would help a lot!

Response: Relevant references 19-20-21 have been added to support the explanation of “air cushion” theory. In these papers, it is mentioned that the micro-structure captures air, thus forming the air cushion effect. At present, we haven't done any research on the comparison between the wettability of textural and non-textural surfaces. So we have no ability to answer your comment. We are very sorry for the loss of some content caused by the problems in uploading files. The image of water droplets on the surface has been re-added in figures 4 and 5, and all the pictures have been re-added with scale bar and related letter. Because the volume of the droplet is 5µl, its radius is 1061 µm, and the scale of figure 4 is 200 µm, even if the contact angle is 138 °, the droplet will cover all the SEM images presented.   Therefore, it is difficult to add droplet contours to the SEM diagram.

2. I was also surprised by the absence of comparisons to any EDM machined surfaces without the magnetic stirring. The sole comparison is brief and done via the citation of only one reference: ref [17]. To make the point clear, the authors should have done and should have presented some tests without magnetic stirring, not necessarily in all conditions for I and T, but enough of them to make the point.

Response: A comparative experiment of adding non-magnetic stirring has been carried out in section 3.1. Two groups of comparative experiments were carried out. The experimental parameters were 1.5 A and 60 µs, 9 A and 60 µs, respectively, and it is found that there are only very few or no micropores on their surfaces, which further proves that magnetic stirring can increase the number of micropores on their surfaces.

 3. A figure presenting a surface taken from SEM cannot tell anything about the water contact angle. We have to wait until the end of the paper to have values of water contact angles, which by the way are not given with their standard deviation. Idem regarding the roughness, it is essential to the study and only given at the end, again without any standard deviation.

Response: Once again, we apologize for the loss of some content and pictures due to file upload problems. Now we have re-added the water contact angle image corresponding to each SEM image in figure 4 and figure 5. As for the standard deviation of contact angle and roughness, we have re-marked it in Figure 8. The viewpoint that the contact angle must increase with the increase of roughness has not been proved in this study, because we find that the contact angle of surface with large roughness is not necessarily large, and the contact angle of surface with small roughness is not necessarily small. Therefore, we believe that the surface morphology has the greatest influence on contact angle, and the effect of roughness on contact angle is not the focus of this study.

4. Figures must have a scale so the reader can have an idea of the lateral size of the features presented and discussed. All images in the figures must have the letter that is associated to them in the caption (a, b, c, d, e, etc…) written on them or below so the reader can associate them easily.

Response: We are very sorry about the loss of some content in file upload. Now we have completed the scale and alphabetical heading sorting on all the pictures.

5. In many places, the authors say the pores are shallow, bulges are bigger, etc… however, base on one SEM image it is not possible for the reader to be sure it is the case as SEM images and the reading we have from them is highly dependant on the contrast and brightness. Profiles of the surface must be presented to illustrate the discussion.

Response: We have added section 3.4 to the manuscript. This section is about the section map. It can be clearly seen from figure 7 (b) that the section of the bulge is higher and the section of the pit is lower.

6. Regarding the number of pores, it would be better if some semi-quantitative numbers are given such as numbers of pores per a certain unit area. Authors are only employing expression such as “certain amount of shallow pores”.

Response: We have revised the term "certain amount of shallow pores" into a more quantitative one “ it is found that when the pulse duration is 15 µs, it was found by SEM that there were 14 holes on its surface.”

7. As there are no scales on images, the reader have no idea of the relevancy of the magnification used to illustrate the results and to support the discussion.

Response: We are very sorry about the loss of some content in file upload. Now we have completed the scale on all the pictures.

8. Clear definition of what the authors define as pores, bulges, craters, coral reef needs to be provided. A simple arrow on one SEM images would be sufficient.

Response: Definitions of "pore", "bulge" and “coral reef” have been given in figure 3.

9. What is the working fluid used in the EDM process?

Response: The working medium used in section 2.1 is a special spark oil. Detailed performance indicators are shown in Table 1.

10. What are the EDS parameters for the chemical composition study?

Response: EDS parameters have been added to section 2.3. The parameters of EDS are kv: 20, mag: 2000, takeoff: 36.1, live times (s): 26.7, and amp time (µs): 7.68.

11. the authors say “The samples were mounted, polished, ultrasonically cleaned, and etched for subsurface microstructure observation by SEM. The etchant consists of 10 ml HF [...]”. I have not seen any subsurface images, nor any discussion directly on data gathered from this subsurface study. If the images from Fig 1 to 4, are those images, then the discussion on the roughness, bulges, pores, etc. on wettability does not make sense because the images would not reflect the real surface. Regarding the HF, there is only fluorine mentioned in the conclusion… and it is in relation with contact angle. That is why I have doubts on the images presented and the relevancy of the discussion (all section 3).

Response: I'm very sorry that some of the content was lost due to the problem of file upload. Now we have re-added it completely. In section 3.4, we have re-added the content on the cross-section layer. See section 3.4 for details.

12. Section 3.1 is highly focused on the gas entering the contact. But what about the cooling pattern/rate due to the circulation of the liquid?

Response: The mode and rate of cooling have been added in section 3.1. The velocity of liquid passing through the processing area is 1.8 m/s, the faster cooling speed prevents gas from overflowing in large quantities from molten metals. So there will be more stomatal characteristics on the surface. Cooling mode is to rely on the non-stop flow of working medium for cooling between the poles.

13. The discussion regarding the Cassie & Baxter theory needs to be developed. Moreover, they focused on rather well defined and regularly patterned surfaces rather than randomly patterned surfaces that are closer to what the authors present.

Response: This Cassie & Baxter theory has been deleted from this manuscript.

14. “This is because there are some shallow pores on their surfaces, which results in droplets easily entering the shallow pores on this type of surface due to gravity and capillary forces.” References are needed to support such a claim.

Response: Document 22 has been added to support the claim that droplets enter pits due to capillary force and gravity.

15.“Therefore, the contact area between the droplet and the solid increases, and it is impossible to catch enough air to form the pressure difference, so there is no "air cushion" effect.” References are needed to support such a claim, or extend the discussion to prove it further.

Response: Documents 19-20-21 have been added to support the argument that the contact area increases without the formation of "air cushion effect".

16. What do the authors mean by “disorderly overlapping”?

Response: The protrusion does not form a certain regularity, but is randomly condensed on the surface. Of course, this is a comparison with (a) (b) (c) in figure 3.

17. The authors selected only two surfaces to conduct elemental analysis. I would be cautious on that as wettability is impacted by a combination of things: cleanliness, material microstructure and composition, surface morphology, size of the drop, etc. The authors must give precisions on where the analysis is performed. Are the spectrum presented in Fig5 obtained from maps, single point analysis, line profiles analysis, etc.? If it is single points, authors must show the surface and where the spectrum is taken. It is extremely important because the surface is heterogeneous in morphology and can consequently be heterogeneous in its composition as well.

Response: The analysis of EDS is based on an area analysis. The measurement area of EDS has been marked in (c) and (h) of figure 3. The fundamental reason for choosing only two samples with the largest contact angle for analysis is that the surface topography of the two samples differs greatly, so we want to explore the differences of elements with different characteristic surfaces.

18. On the spectrum in Fig5, why are the % presented for Ni are based on the L ray and not the K ray?

Response: We contacted the equipment operator and re-measured it. It was found that the content of Ni under K-ray decreased slightly. The results of the retest are shown in figure 6.

19. This is due to Ni has lower evaporation temperature than Ti, and as the pulse duration increases, more Ni element is taken away from the surface by the liquid dielectric, so the content of Ni decrease.” Okay, but what about the possibility of creating a thicker layer of mixed liquid and metal residue that would consequently lower the concentration of Ni.

Response: Based on your point of view, we have carried out XRD analysis of these two samples and found that there are TiC substances on their surfaces. See section 3.3 for details. Therefore, the reasons for the decrease of Ni content in Section 3.3 are further supplemented. The complementary sentence is "And TiC substances are found on the surface of figure 6(c) and (d), which will further decrease the content of Ni".

Reviewer 4 Report

This paper is new, original and well organized.  English language is good in the paper and all references are adequate. Also all parts of paper are important and conclusion is fine. I recommend this paper for publication, but with minor revision:

1) There is an abbreviation "EDM" in the title and abstract of the article. However, this abbreviation is not generally accepted, therefore it is necessary to give the full name of the method in the title of the article and in the abstract.

2) The "Introduction" section describes the Ni-Ti alloy. However, only the section "Equipment and Material" provides information on the exact chemical composition of the alloy. It is necessary at the beginning of the article to give a short description of the composition of the alloy.

3) Line 84 refers to Figure 2. Then a lengthy description of Figure 2 is provided. However, Figure 2 itself is only on line 151. It is necessary to provide Figure 2 immediately after its first mention.

4) In figures 2 and 3 there are no letters a, b, c and etc.

5) Figures 2, 3 and 4 do not show the scale.

Author Response

 Thank you very much for your comments and suggestion.

1. There is an abbreviation "EDM" in the title and abstract of the article. However, this abbreviation is not generally accepted, therefore it is necessary to give the full name of the method in the title of the article and in the abstract.

Response: The full name of EDM, Electric Discharge Machining, has been given in the title and abstract.

2. The "Introduction" section describes the Ni-Ti alloy. However, only the section "Equipment and Material" provides information on the exact chemical composition of the alloy. It is necessary at the beginning of the article to give a short description of the composition of the alloy.

Response: A brief introduction to the composition of nickel-titanium alloys has been added at the beginning of the preamble, which reads "The element composition of medical Ni-Ti alloys is generally that the difference between nickel and titanium element content is not much different, and only contains trace impurities.".

3. Line 84 refers to Figure 2. Then a lengthy description of Figure 2 is provided. However, Figure 2 itself is only on line 151. It is necessary to provide Figure 2 immediately after its first mention.

Response: What I want to introduce at the beginning is the formation mechanism of the pore, that is, the gas entering the molten pool and the source of the gas. To avoid misunderstanding, we will rewrite the opening sentence to "The surface micrographs of magnetic mixed EDMed Ni-Ti alloy under different processing parameters are obviously different. The surface contain different number of pores under most processing parameters". The introduction of pictures began to appear in the later paragraph.

4.  In figures 2 and 3 there are no letters a, b, c and etc.

Response: We are very sorry for the loss of some content caused by the problems in uploading files. Now we have refilled the sorting letters of all the pictures.

5.  Figures 2, 3 and 4 do not show the scale.

Response: We have added scales to all the images.

Round 2

Reviewer 3 Report

The authors considered the comments with good care. The addition of water drops as inset on SEM images helps a lot in linking surface morphology to hydrophobicity. The XRD and section 3.4 are also really good additions to the paper. I would suggest minor revision based on the following suggestions:

Regarding the number of pores on the images, in the first review I was suggesting the authors might report a number of pores by unit area, meaning that the 14 holes observed on the SEM image might correspond to 320 holes per square millimetres. It would be relevant if a mean value calculated over 5 images was reported. If the authors are not able to provide such numbers, reporting only one value (14) does not bring significant information and keeping qualitative comparisons is sufficient, not the best, but sufficient.

A line profile would strengthen the paper, especially in the case of comparison such as in lines 158-159: “The shallow pores on the surface are removed considerably, and the depth of the pores on the surface

159 is deeper than that of Figure 3(h)…” Can the author extract two line profiles from the roughness measurements to support the comparison?

Please add labels on fig 7 to show where the recast layer, the bulge, and the coral reef are.

In the conclusion, I still do not understand why the authors added “without fluoride modification” (line 333). There is no discussion on the effect of fluoride modification in the main body of the paper, nor any quick comparison such as “we obtained …° contact angle while such a value is often obtained with fluoride modification [ref]. I would suggest removing it from the conclusion.

Author Response

Dear  Reviewer:

Thank you for your letter and for the reviewers’ comments concerning our manuscript entitled “Surface Characteristics and Hydrophobicity Analysis of Ni-Ti Alloy by Magnetic Mixed EDM” (ID: 412318). Those comments are all valuable and very helpful for revising and improving our paper, as well as the important guiding significance to our researches. We have studied comments carefully and have made correction which we hope meet with approval. The main corrections in the paper and the responds to the reviewer’s comments are as flowing:

Responds to the reviewers’s comments:

Reviewer:

1. Regarding the number of pores on the images, in the first review I was suggesting the authors might report a number of pores by unit area, meaning that the 14 holes observed on the SEM image might correspond to 320 holes per square millimetres. It would be relevant if a mean value calculated over 5 images was reported. If the authors are not able to provide such numbers, reporting only one value (14) does not bring significant information and keeping qualitative comparisons is sufficient, not the best, but sufficient.

Response: Thank you very much for your valuable comments, because the number and distribution of the holes are not uniform, so we choose the value of 14 to describe the number of holes on high multiple SEM image.

2. A line profile would strengthen the paper, especially in the case of comparison such as in lines 158-159:The shallow pores on the surface are removed considerably, and the depth of the pores on the surface.

2.1. A line profile would strengthen the paper, especially in the case of comparison such as in lines 158-159: The shallow pores on the surface are removed considerably, and the depth of the pores on the surface?

Response: We answered 2 together with 2.1. In fact, we have thought about this problem before, but because our roughness measurement equipment can only provide the roughness values, can not give the measured contours and three-dimensional topography, so I am sorry that further information about this can not be provided.

3. Please add labels on fig 7 to show where the recast layer, the bulge, and the coral reef are.

Response: We have added the markings of the recast layer, bulge and coral reef in Figure 7.

    4. In the conclusion, I still do not understand why the authors added without fluoride modification(line 333). There is no discussion on the effect of fluoride modification in the main body of the paper, nor any quick comparison such as we obtained …° contact angle while such a value is often obtained with fluoride modification [ref]. I would suggest removing it from the conclusion.

Response: In fact, we first consulted the literature and found that some of the ways to obtain high hydrophobic surface were to modify it with fluoride with low surface energy. However, after your opinion, we found that this description is not appropriate in this paper. Therefore, in accordance with your opinion, we deleted "without modification of fluoride" in the conclusion and abstract.